# Associations between lung function and physical and cognitive health in the Canadian Longitudinal Study on Aging (CLSA): A cross-sectional study from a multicenter national cohort

MyLinh Duong[1]*, Ali Usman[2], Jinhui Ma[2], Yangqing Xie[3], Julie Huang[4], Michele Zaman[5], Alex Dragoman[6], Steven Jiatong Chen[6], Malik Farooqi[1], Parminder Raina[2]

1 Firestone Institute for Respiratory Health, Department of Medicine, Division of Respirology, McMaster University, Hamilton, Canada, 2 Department of Health Research Methods, Evidence and Impact, McMaster University, Hamilton, Canada, 3 State Key Laboratory of Respiratory Disease, National Clinical Research Center for Respiratory Disease, Guangzhou Institute of Respiratory Health, Guangzhou Medical University, Guangzhou, China, 4 Lakeridge Health Oshawa, Canada, 5 Department of Epidemiology, Biostatistics and Occupational Health, McGill University, Montreal, Canada, 6 Michael G. DeGroote School of Medicine, McMaster University, Hamilton, Canada

* duongmy@mcmaster.ca

**Data Availability Statement:** Data are available from the Canadian Longitudinal Study on Aging portal (www.clsa-elcv.ca) for researchers who

## Abstract

### Background

Low lung function is associated with high mortality and adverse cardiopulmonary outcomes. Less is known of its association with broader health indices such as self-reported respiratory symptoms, perceived general health, and cognitive and physical performance. The present study seeks to address the association between forced expiratory volume in 1 second ($FEV_1$), an indicator of lung function, with broad markers of general health, relevant to aging trajectory in the general population.

### Methods and findings

From the Canadian general population, 22,822 adults (58% females, mean age 58.8 years [standard deviation (SD) 9.6]) were enrolled from the community between June 2012 and April 2015 from 11 Canadian cities and 7 provinces. Mixed effects regression was used to assess the cross-sectional relationship between $FEV_1$ with self-reported respiratory symptoms, perceived poor general health, and cognitive and physical performance. All associations were adjusted for age, sex, body mass index (BMI), education, smoking status, and self-reported comorbidities and expressed as adjusted odds ratios (aORs). Based on the Global Lung Function Initiative (GLI) reference values, 38% ($n = 8,626$) had normal $FEV_1$ (z-scores >0), 37% ($n = 8,514$) mild (z-score 0 to > −1 SD), 19% ($n = 4,353$) moderate (z-score −1 to > −2 SD), and 6% ($n = 1,329$) severely low $FEV_1$ (z-score = < −2 SD). There was a graded association between lower $FEV_1$ with higher aOR [95% CI] of self-reported

meet the criteria for access to de-identified CLSA data. The CLSA Data and Sample Access Committee is responsible for the review data access applications. Each application is reviewed for its relevance and feasibility. Applications for data access are available through an online process following this link: https://www.clsa-elcv.ca/data-access/data-access-resources.

**Funding:** The author(s) of this research received no specific funding for this work.

**Competing interests:** The authors have declared that no competing interests exist.

**Abbreviations:** AIC, Akaike information criterion; AO, airflow obstruction; aOR, adjusted odds ratio; ATS/ERS, American Thoracic and European Respiratory Society; BIC, Bayesian information criterion; BMI, body mass index; CLSA, Canadian Longitudinal Study on Aging; COPD, chronic obstructive pulmonary disease; CVD, cardiovascular disease; DCS, data collection site; FEV1, forced expiratory volume in 1 second; FVC, forced vital capacity; GLI, Global Lung Function Initiative; LLN, lower limit of normal; OR, odds ratio; PASE, Physical Activity Scale for the Elderly; SD, standard deviation; STROBE, Strengthening the Reporting of Observational Studies in Epidemiology; TUG, Timed Up and Go.

moderate to severe respiratory symptoms (mild $FEV_1$ 1.09 [0.99 to 1.20] $p = 0.08$, moderate 1.45 [1.28 to 1.63] $p < 0.001$, and severe 2.67 [2.21 to 3.23] $p < 0.001$]), perceived poor health (mild 1.07 [0.9 to 1.27] $p = 0.45$, moderate 1.48 [1.24 to 1.78] $p = <0.001$, and severe 1.82 [1.42 to 2.33] $p < 0.001$]), and impaired cognitive performance (mild 1.03 [0.95 to 1.12] $p = 0.41$, moderate 1.16 [1.04 to 1.28] $p < 0.001$, and severe 1.40 [1.19 to 1.64] $p < 0.001$]). Similar graded association was observed between lower $FEV_1$ with lower physical performance on gait speed, Timed Up and Go (TUG) test, standing balance, and handgrip strength. These associations were consistent across different strata by age, sex, tobacco smoking, obstructive, and nonobstructive impairment on spirometry. A limitation of the current study is the observational nature of these findings and that causality cannot be inferred.

## Conclusions

We observed graded associations between lower $FEV_1$ with higher odds of disabling respiratory symptoms, perceived poor general health, and lower cognitive and physical performance. These findings support the broader implications of measured lung function on general health and aging trajectory.

## Author summary

### Why was this study done?

- Lung capacity is a simple measurable marker of lung health and has been strongly linked to higher risks of death and worse heart and lung outcomes. Little is known of the broader association between lung function with general health and cognitive and physical function.

### What did the researchers do and find?

- From a large general Canadian cohort study, 22,822 adults (52% females, average age 58.8 years) from 11 Canadian cities and 7 provinces completed a health survey, spirometry, and cognitive and physical performance testing.

- We observed a graded relationship between lower lung function with higher likelihood of reporting moderate to severe lung symptoms, poor general health rating, and low cognitive performance. Similar graded relationship was also seen between lower lung function with lower performance on mobility, balance, and strength. These associations exist throughout the range of lung function values even at mild levels considered within the normal range. Furthermore, they were consistent across groups of different ages, sex, smoking status, and different patterns of lung function impairment.

**What do these findings mean?**

- These findings suggest that lung capacity may provide important information on general health and cognitive and physical outcomes in the general population.

## Introduction

Pulmonary function measurements expressed as the forced expiratory volume in 1 second ($FEV_1$) or forced vital capacity (FVC) significantly predicts all-cause and cardiovascular mortality. This has been consistently shown in numerous epidemiological studies and across populations of diverse ethnic, geographic, and socioeconomic backgrounds [1–7]. Low $FEV_1$ is also significantly associated with noncardiopulmonary comorbidities including diabetes, chronic kidney diseases, osteoporosis, and dementia in the general population [8–11]. This is independent of tobacco smoking, age, chronic lung diseases, and other comorbidities [2,5]. Due to these strong and consistent associations, it has been suggested that pulmonary function may be a marker of general physiological health and closely relate to the processes of aging [12–15].

Aging is associated with a gradual decline in physiological and functional capacity, which affects all tissues, organs, and systems in a nonuniform way [15]. Furthermore, the decline in physiological and functional capacity is a common risk factor for many chronic noncommunicable diseases and confers high morbidity and mortality [14]. A notable and universal feature of aging is the progressive and generalized dysfunction of the musculoskeletal system leading to reduced muscle mass, strength, and endurance [16]. In its severest form, generalized musculoskeletal dysfunction is associated with significantly higher risks for disability, falls, fractures, hospitalizations, and mortality [17, 18]. While there are many chronic comorbidities including pulmonary diseases that can exacerbate dysfunction of the musculoskeletal system and functional impairment, we speculate that impaired lung function may also be a feature of the primary and generalized process of functional decline associated with aging. In the present study, we seek to understand the relationship between low $FEV_1$ with muscle strength, physical performance, and self-reported health measures independent of lung disease and whether these relationships may be modified by age and other similar risk factors.

The Canadian Longitudinal Study on Aging (CLSA) is an ongoing interdisciplinary cohort study that aims to study the predictors and consequences of aging in a random sample of adults from the Canadian population [19]. In the present study, we examined the cross-sectional baseline data, for associations between $FEV_1$, with self-reported respiratory symptoms, self-perceived poor general health, and cognitive and physical performance. The findings will help to understand the burden and broader implications of low pulmonary function in the general population independent of lung disease. It can also inform on potential novel pathways that can lead to improved lung health and reduce the burden of symptoms and cognitive and physical impairment as the population ages.

## Methods

A protocol of the planned analysis (S1 Protocol) was submitted to the CLSA Data and Sample Access Committee and Hamilton Health Sciences Ethics Committee for approval prior to accessing the data and analysis. CLSA is a large, nationally representative, stratified random sample of 51,338 participants aged 45 to 85 years old at baseline. The study design and

methodology has been published [19]. Enrollment was limited to participants who speak and read English or French. Residents from the Canadian 3 territories, remote geographical regions, First Nations reserves, long-term care facilities, and members of the Armed Forces were excluded. A subset of the CLSA cohort ($n = 30,097$) was randomly selected from 25- to 50-km radius across 11 centers and 7 Canadian provinces (Victoria, Vancouver mainland, Calgary, Winnipeg, Hamilton, Ottawa, Montreal, Sherbrooke, Halifax, and St John's) to attend a data collection site (DCS) for more comprehensive assessments. At these dedicated DCS, participants were interviewed and underwent standardized physical, cognitive, and clinical assessments (comprehensive cohort) to provide data on demographics, lifestyle, health, and clinical information. In the remaining participants (tracking cohort, $n = 21,241$), similar data were collected by a telephone interview. The demographics of the tracking and comprehensive cohorts are provided in S1 Table, which showed comparable baseline characteristics. For the present study, only participants from the comprehensive cohort, with complete baseline data and high-quality spirometry, were included. Selection of high-quality spirometry data was in accordance with the American Thoracic and European Respiratory Society (ATS/ERS) quality standards, which required 3 acceptable maximal efforts and a reproducibility of <150 cc between the 2 highest $FEV_1$ and FVC [20]. The protocol and conduct of CLSA study were approved by the Canadian Institute of Health Research Advisory Committee on Ethical, Legal and Social Issues, Hamilton Research Ethics Board, and all institutional research ethics board of participating sites. All participants provided informed written consent to partcipate in CLSA the study. This study is reported as per the Strengthening the Reporting of Observational Studies in Epidemiology (STROBE) guideline (S1 STROBE Checklist).

## Spirometry measurements

Lung function was measured with the TruFlow Easy-One Air Spirometer (NDD Medical Technologies, Switzerland) and in DCS following a standardized protocol in keeping with ATS/ERS recommendations [20]. Prior to spirometry testing, all participants completed an interviewed-based questionnaire, physical measurements, electrocardiograph, and carotid ultrasound, which took approximately 45 to 60 minutes to complete. During this time, participants did not consume any large meals, alcohol, or cigarettes. Those screened positive for major contraindications to spirometry were excluded (S2 Table) [21]. The highest $FEV_1$ and FVC from 3 acceptable maximal efforts were selected. The Global Lung Function Initiative (GLI) reference values appropriate for age, sex, height, and ethnicity z-scores were used to classify participants into grades of reduced $FEV_1$ [22]. These included (1) normal $FEV_1$ (z-scores >0 standard deviation [SD]); (2) mild (0 to > −1 SD); (3) moderate (−1 SD to > −2 SD); and (4) severe $FEV_1$ (= <−2 SD). The $FEV_1$/FVC GLI lower limit of normal (LLN) was used to identify obstructive impairment. It is important to note that while we have considered all GLI $FEV_1$ z-scores below the population mean (z-score <0 SD) as low, current guidelines considers z-scores >−2 SD to be within the normal range [22].

## Covariates

Self-reported data from questionnaires included age (45 to 54, 55 to 64, 65 to 74, and 75+ years), sex, smoking status (never [lifetime <100 cigarettes], former [last cigarette smoked >12 months], and current), education (primary and below, secondary, and >secondary), known cardiovascular disease (CVD) (angina, congestive heart failure, and myocardial infarction), chronic obstructive pulmonary disease (COPD), asthma, and major chronic diseases (incorporated into the comorbidity index 0, 1 to 2, and >=3). Height and weight were measured with standardized methods and equipment. Body mass index (BMI) was calculated as weight

divided by height-squared and categorized into $<25$, 25 to 30, and $>30$ kg/m$^2$. Self-reported physical activity was assessed by the Physical Activity Scale for the Elderly (PASE) question-naire with higher weighted scores indicating higher activity levels in the previous 7 days [23].

## Outcomes

Self-perceived general health was assessed by asking participants to rate their present heath as either excellent, very good, good, fair, or poor. Responses were reclassified as "POOR" (fair/poor) or "GOOD" (for all else). This self-rating of global health has been extensively studied and shown to be a robust predictor of later health outcomes including mortality [24,25]. Self-reported breathlessness, wheeze, or cough occurring at least 1 night per week or while walking on flat surfaces were classified as moderate to severe respiratory symptoms. Handgrip strength was measured with a dynamometer (Tracker Freedom Wireless), and the highest value from 3 consecutive trials in the dominant hand was recorded [26]. The Timed Up and Go (TUG) test (TUG) was recorded as the time (seconds) to rise from a chair, walk 3 meters at usual pace (with or without walking aids), turn around, walk back, and sit down [27]. Gait speed recorded the speed (meters per second) to walk 4 meters at usual pace [28]. Standing balance recorded the time (seconds) standing on one leg with hands on hips, eyes open, up to a maximum of 60 seconds [29]. All of these physical performance tests have been shown to be strongly predictive of poor long-term health and functional outcomes including mortality [30]. The semantic flu-ency test assessed cognitive performance by asking participants to name as many animals within 60 seconds. Test scores were standardized for age, sex, and education, with scores $<45$ showing significant associations with low self-rated health, mental health, activities of daily liv-ing, and psychiatric disorders [31,32].

## Analysis

Means (SD) and frequency (%) statistics were used to summarize normally distributed contin-uous variables and categorical data, respectively. The assumption of normality and constant variance of the FEV$_1$, FVC, and covariates were assessed by visual inspection of histograms and plots of residuals against fitted values. Multilevel logistic regression was used to estimate the association between low FEV$_1$ severity categories (relative to FEV$_1 > 0$ SD as reference) with categorical outcomes. Similar multilevel linear regression was used to estimate the mean differences in physical performance outcomes for each FEV$_1$ levels relative to the reference group (FEV$_1 > 0$ SD). Unadjusted estimates are provided, and adjusted estimates were calcu-lated controlling for age, sex, BMI, education, smoking status, self-reported asthma, COPD, CVD, and comorbidity index (excluding asthma, COPD, and CVD), with centers as random effect. The goodness of fit tests (likelihood ratio test, deviance, Akaike information criterion [AIC], and Bayesian information criterion [BIC]), multicollinearity (tolerance and variance inflation factor), and visual inspection of residuals were conducted to assess model stability and robustness. Trimmed inflation and analytical (rescaled) weights were applied to reduce the effect of selection bias and maintain the national representativeness and generalizability of the data [19]. Similar analyses after removing participants with spirometric airflow obstruction (AO; FEV$_1$/FVC$<$LLN) were performed to ensure that our findings were not confounded by diagnosed and undiagnosed COPD. All analyses were performed with STATA 14 (Stata, Texas, USA).

## Results

From the comprehensive cohort, 22,822 participants (52% females, mean age 58.8 [SD 9.6]) with high-quality spirometry and no missing data were included in the study. The baseline

**Table 1. Baseline characteristics by categories of FEV$_1$ levels.**

| | Overall | Categories of FEV$_1$ according to GLI z-scores | | | |
|---|---|---|---|---|---|
| | | >0 SD | 0 to > −1 SD | −1 to > −2 SD | =<−2 SD |
| | | Normal | Mild | Moderate | Severe |
| N, % | 22,822 (100%) | 8,626 (38%) | 8,514 (37%) | 4,353 (19%) | 1,329 (6%) |
| FEV$_1$% | 95% (SD 15.4%) | 109.7% (SD 8.1) | 93% (SD 4.2) | 79.5% (SD 4.8%) | 61.1% (SD 9.1) |
| FEV$_1$/FVC ratio < 0.70 | 2,661 (11%) | 202 (2%) | 654 (7%) | 1,007 (22%) | 798 (58%) |
| <LLN | 1,155 (5.4%) | 24 (0.2%) | 151 (2%) | 392 (10%) | 588 (45%) |
| Female | 11,981 (52%) | 4,629 (53%) | 4,494 (52%) | 2,193 (49%) | 665 (49%) |
| Age, years 45 to 54 | 6,235 (44%) | 2,351 (44%) | 2,306 (44%) | 1,209 (45%) | 369 (44%) |
| 55 to 64 | 7,769 (30%) | 2,934 (30%) | 2,937 (31%) | 1,488 (30%) | 410 (27%) |
| 65 to 74 | 5,396 (16%) | 2,079 (17%) | 2,011 (17%) | 977 (15%) | 329 (17%) |
| 75+ | 3,422 (10%) | 1,262 (9%) | 1,260 (9%) | 679 (10%) | 221 (11%) |
| Height, m | 1.69 (SD 0.1) | 1.69 (SD 0.1) | 1.69 (SD 0.1) | 1.69 (SD 0.1) | 1.69 (SD 0.1) |
| BMI, kgm$^{-2}$ <25 | 7,082 (33%) | 3,084 (38%) | 2,486 (32%) | 1,150 (29%) | 362 (30%) |
| 25 to 30 | 9,226 (40%) | 3,671 (42%) | 3,491 (41%) | 1,620 (37%) | 444 (32%) |
| >30 | 6,496 (26%) | 1,866 (20%) | 2,527 (27%) | 1,580 (34%) | 523 (38%) |
| Education primary | 1,074 (4%) | 309 (3%) | 402 (4%) | 248 (5%) | 115 (8%) |
| Secondary/trade | 2,114 (9%) | 744 (8%) | 780 (8%) | 440 (10%) | 150 (11%) |
| University | 19,602 (87%) | 7,561 (89%) | 7,323 (87%) | 3,659 (86%) | 1,059 (80%) |
| Smoking never | 7,318 (34%) | 2,933 (36%) | 2,789 (35%) | 1,288 (32%) | 308 (26%) |
| Former | 13,515 (57%) | 5,238 (59%) | 5,033 (57%) | 2,524 (56%) | 720 (51%) |
| Current | 1,851 (8%) | 395 (5%) | 641 (8%) | 518 (12%) | 297 (22%) |
| Physical activity | 153.2 (SD 76.6) | 158.7 (SD 75.8) | 152.3 (SD 75.7) | 147.9 (SD 77.6) | 139.2 (SD 81.8) |
| COPD | 1,040 (4%) | 194 (2%) | 306 (3%) | 289 (6%) | 251 (16%) |
| Asthma | 2,940 (13%) | 727 (8%) | 1,064 (13%) | 776 (18%) | 373 (29%) |
| CVD | 2,593 (9%) | 750 (7%) | 932 (9%) | 647 (12%) | 264 (17%) |
| No chronic conditions | 3,297 (10%) | 1,409 (22%) | 1,231 (20%) | 533 (18%) | 124 (13%) |
| >= 3 chronic conditions | 8,664 (35%) | 3,043 (33%) | 3,203 (35%) | 1,825 (39%) | 593 (45%) |

Data are provided as counts and % of total within each FEV$_1$ category/column or as means and SDs for continuous variables. FEV$_1$ z-scores were calculated using the GLI 2012 predicted values appropriate for age, sex, height, and ethnicity.

N = sample size within each FEV$_1$ category.

Physical activity was self-reported for the previous 7 days using the PASE instrument with higher scores indicating higher physical activity. Low physical activity was defined as achieving less than 150 minutes per week of moderate intensity activity. Asthma, COPD, CVD, and chronic conditions were self-reported at baseline.

BMI, body mass index calculated as weight (kg) divided by height (m) squared; COPD, chronic obstructive pulmonary disease; CVD, cardiovascular disease; FEV$_1$, forced expiratory volume in 1 second; FVC, forced vital capacity; GLI, Global Lung function Initiative; LLN, lower limits of normal from GLI predicted norms for age, height, sex, and ethnicity; PASE, Physical Activity Scale for the Elderly; SD, standard deviation.

characteristics of included participants are provided in Table 1. Among this cohort, 38% (n = 8,626) had normal FEV$_1$, 37% (n = 8,514) mild, 19% (n = 4,353) moderate, and 6% (n = 1,329) severely low FEV$_1$ (=<−2 SD). The overall prevalence of AO defined by FEV$_1$/FVC<0.70 was 11% (n = 2,661) and by GLI FEV$_1$/FVC <LLN was 5.4% (n = 1,155). The prevalence of AO increased with increasing level of FEV$_1$ impairment, reaching as high as 58% (by the FEV$_1$/FVC < 0.7 criterion) and 45% (by the GLI FEV$_1$/FVC <LLN) in the severely low FEV$_1$ category. Compared to the overall cohort, severely low FEV$_1$ had higher percentages of current smokers (8% versus 22%), BMI >30 (26% versus 38%), lower education level (4% versus 8%), and lower mean physical activity (153.2 ± 76.6 versus 139.2 ± 81.8). There were also

higher percentages of self-reported asthma (13% versus 29%), COPD (4% versus 16%), CVD (9% versus 17%), and multiple comorbidities (35% versus 45%).

## Self-perceived poor general health, respiratory symptoms, cognitive impairment, and $FEV_1$

The proportion of the overall cohort, reporting mild-moderate respiratory symptoms, was approximately 24% ($n = 5,367$), perceived poor health 8% ($n = 1,736$), and impaired cognitive performance 30% ($n = 6,684$) (Table 2). The prevalence, unadjusted odds ratios (ORs), and adjusted odds ratios (aORs) for all 3 outcomes showed a graded increase with lower $FEV_1$ (Table 2, Fig 1). For perceived poor health, the unadjusted OR across the mild, moderate, and severe $FEV_1$ categories were 1.34 (95% CI 1.16 to 1.54; $p < 0.001$), 2.28 (1.96 to 2.66; $p < 0.001$), and 4.07 (3.35 to 4.94; $p < 0.001$), respectively. After adjusting for differences in demographics between categories, the corresponding aORs were 1.07 (0.9 to 1.27; $p = 0.45$), 1.48 (1.24 to 1.78; $p < 0.001$), and 1.82 (1.42 to 2.33; $p < 0.001$). For self-reported moderate to severe respiratory symptoms, the unadjusted ORs across categories were 1.34 (1.23 to 1.46;

**Table 2. Self-reported perceived poor health, respiratory symptoms, and cognitive impairment for different grades of low $FEV_1$ compared to reference ($FEV_1 > 0$ SD) in the overall cohort and in a subgroup without spirometry AO (shown here as $FEV_1/FVC > = LLN$).**

| | Categories of $FEV_1$ according to GLI z-scores | | | |
| --- | --- | --- | --- | --- |
| | >0 SD (reference) | 0 to > −1 SD | −1 to > −2 SD | =<−2 SD |
| Total 22,822 | Normal 8,626 | Mild 8,514 | Moderate 4,353 | Severe 1,329 |
| **Perceived poor health** | | | | |
| Overall 1,736 (7.6%) | 444 (5%) | 574 (7%) | 489 (11%) | 229 (18%) |
| Unadjusted OR for overall | 1 | 1.34 (1.16, 1.54) $p < 0.001$ | 2.28 (1.96, 2.66) $p < 0.001$ | 4.07 (3.35, 4.94) $p < 0.001$ |
| aOR for overall | 1 | 1.07 (0.9, 1.27) $p = 0.450$ | 1.48 (1.24, 1.78) $p < 0.001$ | 1.82 (1.42, 2.33) $p < 0.001$ |
| Unadjusted OR for $FEV_1/FVC > = LLN$ | 1 | 1.33 (1.15, 1.54) $p < 0.001$ | 2.34 (2.01, 2.74) $p < 0.001$ | 4.50 (3.36, 5.68) $p < 0.001$ |
| aOR for $FEV_1/FVC > = LLN$ | 1 | 1.08 (0.91, 1.28) $p = 0.363$ | 1.60 (1.33, 1.92) $p < 0.001$ | 1.91 (1.42, 2.55) $p < 0.001$ |
| **Moderate to severe symptoms** | | | | |
| Overall 5,367 (24%) | 1,607 (20%) | 1,919 (22%) | 1,246 (29%) | 595 (45%) |
| Unadjusted OR for overall | 1 | 1.34 (1.23, 1.46) $p < 0.001$ | 2.03 (1.83, 2.25) $p < 0.001$ | 5.11 (4.36, 6.00) $p < 0.001$ |
| aOR for overall cohort | 1 | 1.10 (0.98, 1.20) $p = 0.085$ | 1.45 (1.28, 1.63) $p < 0.001$ | 2.67 (2.21, 3.23) $p < 0.001$ |
| Unadjusted OR for $FEV_1/FVC > = LLN$ | 1 | 1.34 (1.23, 1.47) $p < 0.001$ | 2.00 (1.80, 2.22) $p < 0.001$ | 4.92 (4.02, 6.02) $p < 0.001$ |
| aOR for $FEV_1/FVC > = LLN$ | 1 | 1.15 (1.04, 1.26) $p = 0.007$ | 1.56 (1.39, 1.76) $p < 0.001$ | 2.95 (2.33, 3.72) $p < 0.001$ |
| **Cognitive impairment** | | | | |
| Overall 6,684 (30.3%) | 2,365 (28%) | 2,456 (30%) | 1,358 (33%) | 475 (40%) |
| Unadjusted OR for overall | 1 | 1.06 (0.99, 1.15) $p = 0.113$ | 1.19 (1.09, 1.31) $p < 0.001$ | 1.53 (1.33, 1.76) $p < 0.001$ |
| aOR for overall | 1 | 1.03 (0.95, 1.12) $p = 0.414$ | 1.16 (1.04, 1.28) $p = 0.005$ | 1.40 (1.19, 1.64) $p < 0.001$ |
| Unadjusted OR for $FEV_1/FVC > = LLN$ | 1 | 1.06 (0.98, 1.15) $p = 0.126$ | 1.21 (1.11, 1.33) $p < 0.001$ | 1.70 (1.42, 2.03) $p < 0.001$ |
| aOR for $FEV_1/FVC > = LLN$ | 1 | 1.03 (0.95, 1.12) $p = 0.479$ | 1.17 (1.05, 1.29 $p = 0.003$ | 1.44 (1.17, 1.76) $p < 0.001$ |

For each outcome, raw data expressed as frequencies (%) of each $FEV_1$ category/column are provided for the overall cohort in the first row. Unadjusted and aORs relative to reference group ($FEV_1 > 0$ SD) with 95% CI and $p$-values were estimated for different $FEV_1$ categories for the overall cohort and for subgroup excluding spirometric AO (participants with $FEV_1/FVC < LLN$ excluded). aORs were adjusted for age, sex, BMI, smoking status (never, former, and current), education (less than secondary, secondary, and postsecondary), physical activity, self-reported asthma/COPD/CVD, and the number of chronic conditions. Moderate to severe respiratory symptoms refer to breathlessness, cough, or wheeze with walking on flat surfaces or occurring at nighttime at least once per week. Analyses were performed for the overall cohort and separately for the remaining participants ($n = 21,667$) after removing those with GLI $FEV_1/FVC < LLN$.

AO, airflow obstruction; aOR, adjusted odds ratio; BMI, body mass index; COPD, chronic obstructive pulmonary disease; CVD, cardiovascular disease; $FEV_1$, forced expiratory volume in 1 second; FVC, forced vital capacity; GLI, Global Lung function Initiative; LLN, lower limit of normal for age, sex, height, and ethnicity using the GLI reference values; SD, standard deviation.

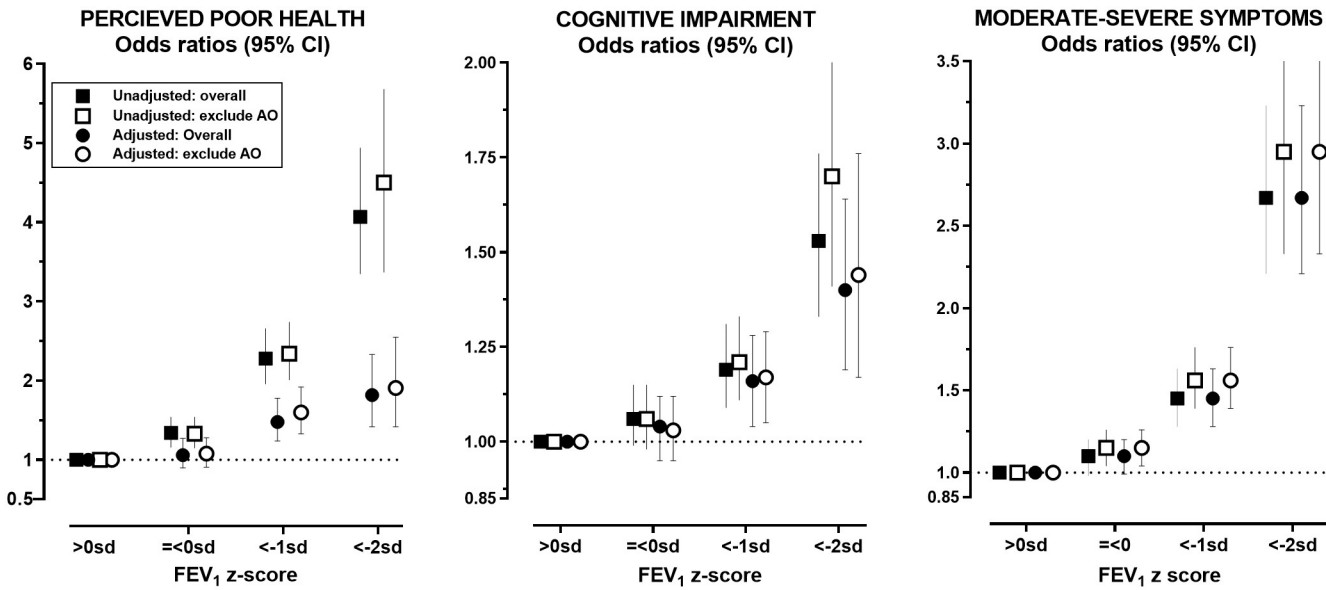

**Fig 1. Unadjusted and aORs (95% CI) for self-reported perceived poor health status, respiratory symptoms, and low cognitive scores by grades of low FEV$_1$ relative to reference group in the overall cohort and in participants without spirometry AO.** AO, airflow obstruction; aOR, adjusted odds ratio; FEV$_1$, forced expiratory volume in 1 second; SD, standard deviation.

$p < 0.001$), 2.03 (1.83 to 2.25; $p < 0.001$), and 5.11 (4.36 to 6.00; $p < 0.001$), with corresponding aORs of 1.10 (0.98 to 1.12; $p = 0.41$), 1.45 (1.28 to 1.63; $p < 0.001$), and 2.67 (2.21 to 3.23; $p < 0.001$). Similar trend was observed for impaired cognitive performance, with unadjusted ORs of 1.06 (0.99 to 1.15; $p = 0.113$), 1.19 (1.09 to 1.31; $p < 0.001$), and 1.53 (1.33 to 1.76; $p < 0.001$) across increasing lower FEV$_1$ categories. The corresponding aORs for impaired cognitive performance were 1.03 (0.95 to 1.12; $p = 0.41$), 1.16 (1.04 to 1.28; $p = 0.005$), and 1.40 (1.19 to 1.64; $p < 0.001$). While the ORs for severe FEV$_1$ were the highest for all outcomes, the absolute numbers of affected participants in the mild and moderate FEV$_1$ categories combined exceeded the numbers with severe FEV$_1$. For example, 1,063, 3,165, and 3,814 participants in the mild and moderate FEV$_1$ groups combined, reported poor general health, moderate to severe respiratory symptoms, and impaired cognitive performance, respectively. These numbers were 5- to 8-folds higher than the 229 (perceived poor health), 595 (moderate to severe respiratory symptoms), and 475 (impaired cognitive performance) participants in the severe FEV$_1$ categories.

ORs and 95% CIs are presented as adjusted (circle symbols) and unadjusted (square symbols) estimates relative to reference group (FEV$_1$ z score >0 SD). For adjusted multilevel logistic regression model, see Methods section. ORs were calculated for the overall cohort (closed) and after removing participants with spirometric AO (AO = FEV$_1$/FVC <LLN) (open). $p$-Values for comparisons are provided in Table 2.

## FEV$_1$ and physical performance

There were similar trends observed between lower FEV$_1$ with declining physical performances on the TUG, gait speed, standing balance, and handgrip strength (Table 3, Fig 2). Compared to normal FEV$_1$, the unadjusted mean difference in gait speed for mild, moderate, and severe FEV$_1$ were −0.011 m/s (95% CI −0.017, −0.004; $p = 0.001$), −0.034 (−0.042, −0.026; $p < 0.001$), and −0.074 (−0.087, −0.062; $p < 0.001$), respectively. These corresponded to adjusted mean differences of −0.002 m/s (−0.008, 0.004; $p = 0.54$), −0.018 (−0.026, −0.009; $p < 0.001$), and

**Table 3. Mean differences (unadjusted and adjusted) in physical performance by different grades of low FEV$_1$ compared to reference group (FEV$_1 > 0$ SD) for the overall cohort and subgroup without AO on spirometry (shown here as FEV$_1$/FVC $> =$ LLN).**

| | | Categories of FEV$_1$ according to GLI z-scores | | |
|---|---|---|---|---|
| **OVERALL** | **>0 SD (ref)** | **0 to > −1 SD** | **−1 to > −2 SD** | **=<−2 SD** |
| Total 22,822 | Normal 8,626 | Mild 8,514 | Moderate 4,353 | Severe 1,329 |
| **Gait speed, m/s** | | | | |
| Mean 1.01 (SD 0.19) | 1.02 (0.18) | 1.01 (0.19) | 0.99 (0.19) | 0.95 (0.20) |
| Unadjusted: overall | 0 | −0.011 (−0.017, −0.004) $p = 0.001$ | −0.034 (−0.042, −0.026) $p < 0.001$ | −0.074 (−0.087, −0.062) $p < 0.001$ |
| Adjusted: overall | 0 | −0.002 (−0.008, 0.004) $p = 0.535$ | −0.018 (−0.026, −0.009) $p < 0.001$ | −0.039 (−0.053, −0.026) $p < 0.001$ |
| Unadjusted: FEV$_1$/FVC $> =$ LLN | 0 | −0.012 (−0.018, −0.005) $p < 0.001$ | −0.036 (−0.044, −0.028) $p < 0.001$ | −0.086 (−0.103, −0.070) $p < 0.001$ |
| Adjusted: FEV$_1$/FVC $> =$ LLN | 0 | −0.002 (−0.009, 0.004) $p = 0.455$ | −0.018 (−0.027, −0.01) $p < 0.001$ | −0.051 (−0.068, −0.033) $p < 0.001$ |
| **Standing balance, seconds** | | | | |
| Mean 45.4 (SD 21.3) | 47.6 (20) | 45.4 (21.4) | 43.1 (22.3) | 38 (23.8) |
| Unadjusted: Overall | 0 | −2.19 (−2.86, −1.52) $p < 0.001$ | −4.72 (−5.60, −3.85) $p < 0.001$ | −9.40 (−10.93, −7.87) $p < 0.001$ |
| Adjusted: Overall | 0 | −0.97 (−1.59, −0.35) $p = 0.002$ | −2.57 (−3.37, −1.77) $p < 0.001$ | −5.31 (−6.75, −3.87) $p < 0.001$ |
| Unadjusted: FEV$_1$/FVC $> =$ LLN | 0 | −2.22 (−2.89, −1.54) $p < 0.001$ | −5.31 (−6.23, −4.89) $p < 0.001$ | −10.65 (−12.69, −8.61) $p < 0.001$ |
| Adjusted: FEV$_1$/FVC $> =$ LLN | 0 | −0.95 (−1.57, −0.33) $p = 0.003$ | −3.04 (−3.87, −2.20) $p < 0.001$ | −6.78 (−8.72, −4.85) $p < 0.001$ |
| **TUG, seconds** | | | | |
| Mean 9.2 (SD 2.1) | 9.0 (2) | 9.2 (2) | 9.5 (2.4) | 10 (2.8) |
| Unadjusted: Overall | 0 | 0.129 (0.068, 0.190) $p < 0.001$ | 0.471 (0.386, 0.556) $p < 0.001$ | 0.950 (0.763, 1.137) $p < 0.001$ |
| Adjusted: Overall | 0 | 0.03 (−0.03, 0.09) $p = 0.304$ | 0.28 (0.19, 0.36) $p < 0.001$ | 0.50 (0.30, 0.70) $p < 0.001$ |
| Unadjusted: FEV$_1$/FVC $> =$ LLN | 0 | 0.133 (0.072, 0.194) $p < 0.001$ | 0.507 (0.417, 0.596) $p < 0.001$ | 1.168 (0.882, 1.454) $p < 0.001$ |
| Adjusted: FEV$_1$/FVC $> =$ LLN | 0 | 0.03 (−0.031, 0.093) $p = 0.324$ | 0.30 (0.21, 0.39) $p < 0.001$ | 0.71 (0.39, 1.03) $p < 0.001$ |
| **Grip strength, kg** | | | | |
| Mean 37.0 (SD 12.2) | 37.8 (12.2) | 36.7 (12.2) | 36.5 (12.1) | 35.0 (12) |
| Unadjusted: Overall | 0 | −0.95 (−1.39, −0.51) $p < 0.001$ | −1.23 (−1.77, −0.69) $p < 0.001$ | −2.79 (−3.65, −1.93) $p < 0.001$ |
| Adjusted: Overall | 0 | −1.08 (−1.36, −0.81) $p < 0.001$ | −1.88 (−2.23, −1.53) $p < 0.001$ | −3.44 (−4.10, −2.80) $p < 0.001$ |
| Unadjusted: FEV$_1$/FVC $> =$ LLN | 0 | −0.93 (−1.37, −0.48) $p < 0.001$ | −1.22 (−1.78, −0.66) $p < 0.001$ | −2.91 (−4.00, −1.82) $p < 0.001$ |
| Adjusted: FEV$_1$/FVC $> =$ LLN | 0 | −1.10 (−1.37, −0.82) $p < 0.001$ | −1.96 (−2.32, −1.60) $p < 0.001$ | −4.20 (−5.05, −3.34) $p < 0.001$ |

For each outcome, raw data expressed as means (SD) observed for each FEV$_1$ category/column are provided in the first row for the overall cohort. Unadjusted and adjusted mean differences (95% CI) relative to the reference group (FEV$_1 > 0$ SD) were estimated for the different FEV$_1$ category for the overall cohort and after removing participants with AO (GLI FEV$_1$/FVC <LLN). Adjusted estimates were controlled for age, sex, BMI, smoking status (never, former, and current), education (less than secondary, secondary, and postsecondary), physical activity, self-reported asthma/COPD/CVD, and the number of chronic conditions.

AO, airflow obstruction; BMI, body mass index; COPD, chronic obstructive pulmonary disease; CVD, cardiovascular disease;; FVC, forced vital capacity; GLI, Global Lung function Initiative; LLN, lower limit of normal for age, sex, height, and ethnicity using the GLI reference values; SD, standard deviation; TUG, Timed Up and Go.

−0.039 (−0.053, −0.026; $p < 0.001$). Similar graded increase in unadjusted (−2.19 seconds [−2.86, −1.52; $p < 0.001$], −4.72 [−5.60, −3.85; $p < 0.001$], and −9.40 [−10.93, −7.87; $p < 0.001$]) and adjusted (−0.97 seconds [−1.59, −0.35; $p = 0.020$], −2.57 [−3.37, −1.77; $p < 0.001$], and −5.31 [−6.75, −3.87; $p < 0.001$]) mean differences in standing balance were observed with lower FEV$_1$ categories. For TUG, the unadjusted (0.129 seconds [0.068, 0.190; $p < 0.001$], 0.471 [0.386, 0.556; $p < 0.001$], and 0.950 [0.763, 1.137; $p < 0.001$] and adjusted (0.033 seconds [−0.029, 0.094; $p = 0.304$], 0.276 [0.189, 0.362; $p < 0.001$], and 0.503 [0.301, 0.705; $p < 0.001$]) mean differences showed a similar trend with lower FEV$_1$ categories. Last, the unadjusted (−0.95 kg [−1.39, −0.51; $p < 0.001$], −1.23 [−1.77, −0.69; $p < 0.001$], and −2.79 [−3.65, −1.93; $p < 0.001$]) and adjusted (−1.08 kg [−1.36, −0.81; $p < 0.001$], −1.88 [−2.23, −1.53; $p < 0.001$], and −3.44 [−4.10, −2.80; $p < 0.001$]) mean differences in handgrip strength

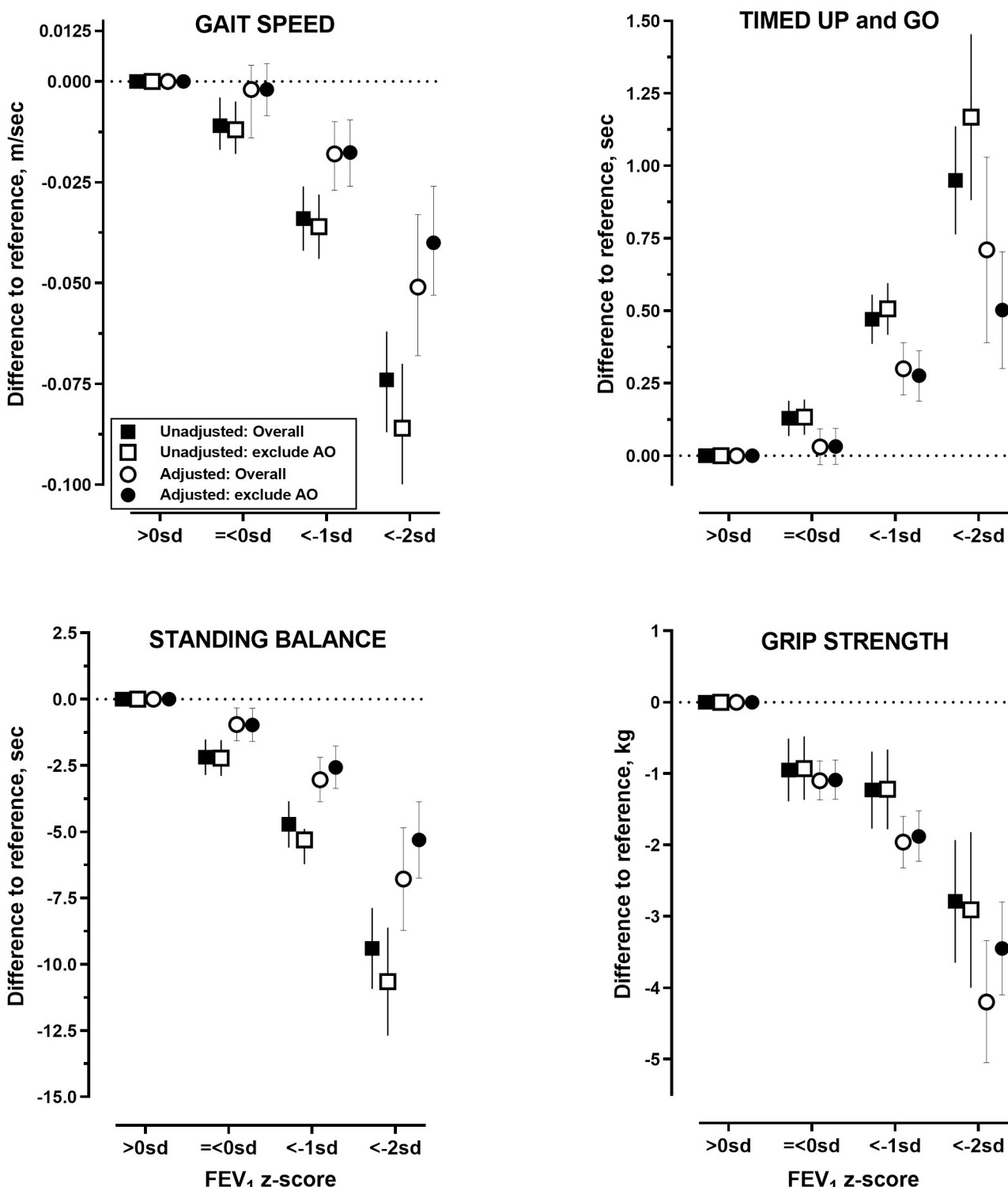

**Fig 2. Unadjusted and adjusted mean differences (95% CI) in physical performance by grades of low FEV$_1$ relative to reference group (FEV$_1$ > 0 SD) in the overall cohort and in participants without AO on spirometry.** AO, airflow obstruction; FEV$_1$, forced expiratory volume in 1 second; SD, standard deviation.

showed a strong and significant increase in effect size with progressively lower $FEV_1$ categories.

Unadjusted (square symbols) mean differences between each level of low $FEV_1$ relative to reference ($FEV_1\% > 0$ SD) for TUG, gait speed, standing balance, and handgrip strength. For the methods used to calculate adjusted estimates (circle symbols), see Methods section. Closed symbols represent data for the overall cohort; open symbols represent data for subgroup after removing participants with spirometry AO (AO = $FEV_1$/FVC <LLN). $p$-Values for comparisons are provided in Table 3.

## Sensitivity analyses

To avoid any confounding by undiagnosed COPD or AO, we conducted 2 sensitivity analyses. First, all analyses were repeated after removing participants with AO using the GLI $FEV_1$/FVC<LLN criterion ($n = 1,155$). This did not materially change the above findings, suggesting the results were independent of AO (Tables 2 and 3). Second, we conducted the analysis using FVC and found similar associations between lower levels of FVC categories with all outcomes (S3 Table), further supporting the robustness and generalizability of these associations to all lung function impairment.

## Stratified analyses by age, sex, smoking status, and COPD/asthma

Unadjusted and adjusted analyses showed similar pattern of graded association between lower $FEV_1$ with higher odds of respiratory symptoms, perceived poor health, impaired cognitive performance (S4 and S5 Tables), and lower physical performance (S6 and S7 Tables) were observed across stratified groups by sex (Fig 3), smoking status (Fig 4), and age (Fig 5).

Multilevel linear regression was used to estimate the differences between each level of $FEV_1$ relative to reference ($FEV_1\% > 0$ SD) on gait speed, standing balance, TUG, and handgrip strength stratified by males (closed symbols) and females (open symbols). All analyses were adjusted for age, BMI, smoking status, education, physical activity, self-reported asthma/COPD/CVD, and number of self-reported chronic noncommunicable conditions. Only $p$-values of $^{\Psi}<0.005$ and $^{\delta}<0.0005$ compared to reference within stratum are reported. Corresponding numerical data can be found in Appendix 7. Unadjusted estimates can be found in Appendix 6.

Multilevel linear regression was used to estimate the differences between each level of $FEV_1$ z-score relative to reference ($FEV_1\% > 0$ SD) on gait speed, standing balance, TUG, and

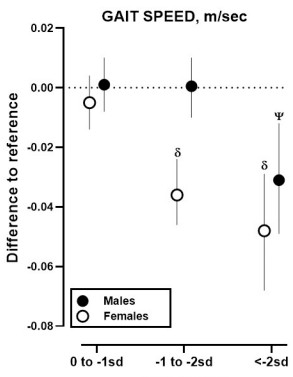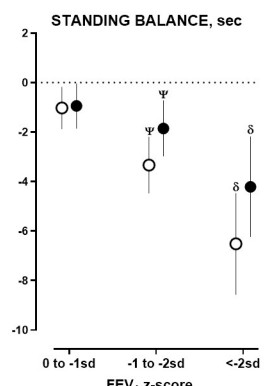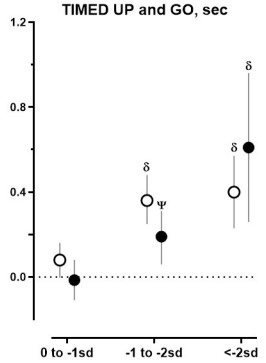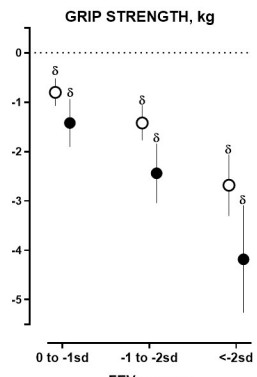

**Fig 3. Associations between physical performances with grades of $FEV_1$ stratified by sex.** $FEV_1$, forced expiratory volume in 1 second; SD, standard deviation.

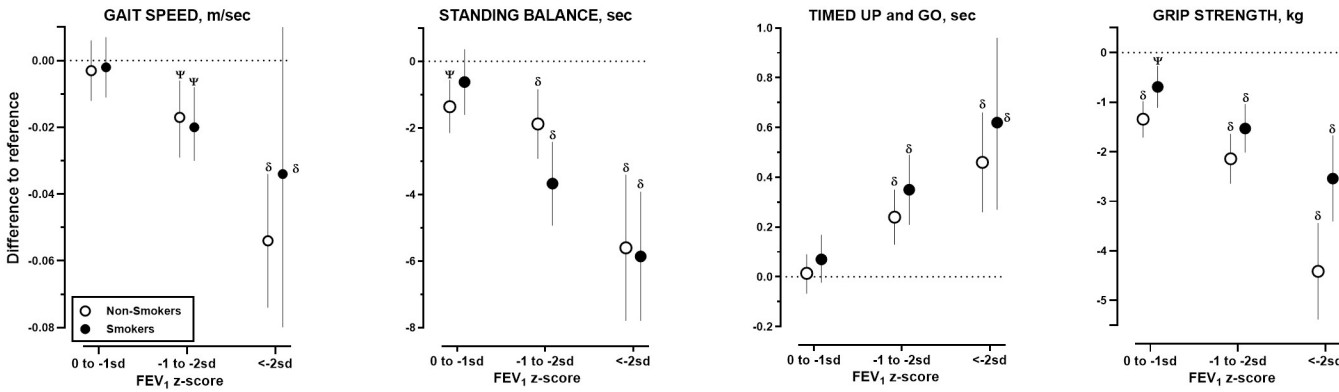

**Fig 4. Associations between physical performances with grades of FEV$_1$ stratified by smoking history.** FEV$_1$, forced expiratory volume in 1 second; SD, standard deviation.

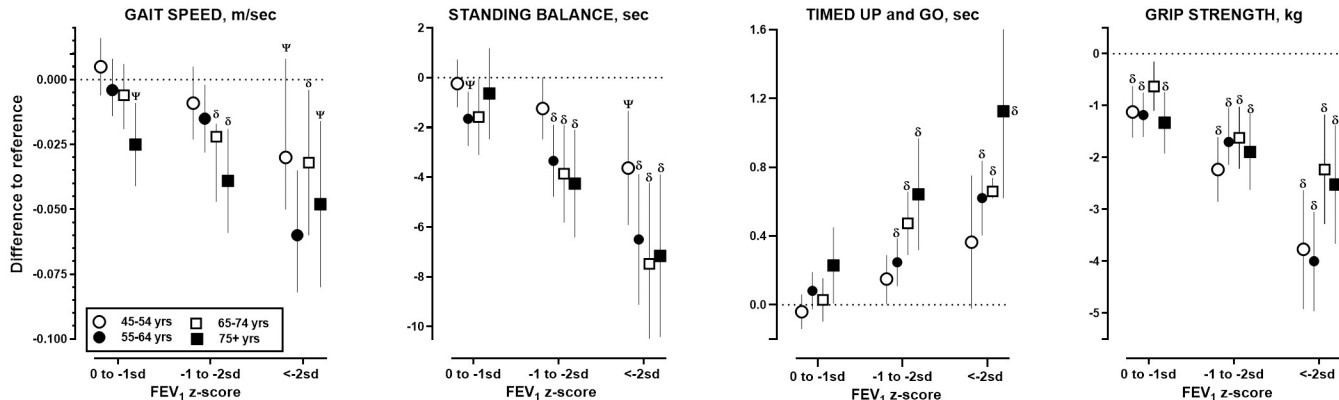

**Fig 5. Associations between physical performances with grades of FEV$_1$ stratified by age.** FEV$_1$, forced expiratory volume in 1 second; SD, standard deviation.

handgrip strength stratified by self-reported smoking history: nonsmokers (open symbols) and smokers (close symbols). All analyses were adjusted for age, sex, BMI, education, physical activity, self-reported asthma/COPD/CVD, and number of self-reported chronic noncommunicable conditions. Only $p$-values of $^{\Psi} < 0.005$ and $^{\delta} < 0.0005$ compared to reference within stratum are reported. Corresponding numerical data can be found in Appendix 7. Unadjusted estimates are provided in Appendix 6.

Multilevel linear regression was used to estimate the differences between each level of FEV$_1$ z-score relative to reference (FEV$_1\% > 0$ SD) on gait speed, standing balance, TUG, and handgrip strength stratified by baseline age categories. All analyses were adjusted for sex, BMI, smoking status, education, physical activity, self-reported asthma/COPD/CVD, and number of self-reported chronic noncommunicable conditions. Only $p$-values of $^{\Psi} < 0.005$ and $^{\delta} < 0.0005$ compared to reference within stratum are reported. Corresponding numerical data can be found in Appendix 7 and adjusted estimates in Appendix 6.

## Discussion

In this large representative sample of the Canadian population aged 45 to 85 years living independently in the community, we found significant and graded associations between higher

odds of perceived poor general health, moderate to severe respiratory symptoms, and impaired cognitive performance with lower $FEV_1$. The pattern of association persisted even after controlling for potential confounders and differences in baseline characteristics between $FEV_1$ severity groups. Similar graded associations were observed between lower $FEV_1$ with lower performance on validated physical assessment tools. These relationships was evident throughout all grades of lower $FEV_1$, even for mildly reduced levels generally regarded as within the limits of normal (i.e., above $-2$ SD). Furthermore, these associations were highly consistent across different age groups, sex, smoking status, and obstructive and nonobstructive impairment on spirometry. These findings suggest that the association between lung function with quality of life and functional measures are robust and generalizable to the wider population, independent of lung disease. It also highlights the potential underestimation of the burden associated with mild to moderately reduced levels of $FEV_1$.

The association between lower lung function with excess mortality in the general population has long been recognized by numerous population-based studies [1–7]. There is emerging evidence for a wider association between low lung function with various chronic nonpulmonary diseases [8–11]. However, there is a paucity of data linking lung function with burden of symptoms, functional, and physical impairment in the wider population without lung diseases. Addressing this gap may help to better understand the burden of low lung function in the general population and offer new insight into pathways that may link low lung function with nonpulmonary comorbidities. The novel finding here is the consistent and graded association between lower $FEV_1$ with higher ORs for self-reported respiratory symptoms, perceived poor health status, cognitive impairment, and lower physical performance. Since these were cross sectional data, the cause–effect implications are not known. These findings, however, suggest that there may be common pathways between reduced lung function with reduced cognitive and physical performance. As the latter outcomes are strongly associated with future risks of disability, falls, hospitalization, and mortality in later years [16–18,33] we propose that reduced lung function even at very mild levels may be an important early indicator of functional impairment as the population age. Therefore, public health strategies, which effectively maintain optimal lung health, may have wider impact and benefits to perceived general health, respiratory symptoms, cognitive and physical functioning, and the overall health trajectory.

A number of previous studies have reported on the high prevalence of physical function impairment and sarcopenia with COPD [16,33–36]. We found that removing participants with obstructive lung function impairment (a cardinal feature of COPD) did not change our findings. Furthermore, we found similar graded and significant associations between FVC levels with these same outcomes. These findings further support the generalizability of these associations to the wider population independent of AO. In fact, the large majority of low $FEV_1$ were nonobstructive impairment and in keeping with the high prevalence of "restrictive" pulmonary impairment previously reported in other populations from high-income countries [37]. Moreover, we found that these associations were present in nonsmokers to the same extent as smokers, supporting their independence from COPD and tobacco smoking.

It is important to note that the effect sizes for low $FEV_1$ for different physical performance measures were only mild to moderate, which is to be expected as the CLSA cohort is a community-based cohort and likely to be relatively healthy at baseline. In addition, we had carefully adjusted for a large number of potential confounders to avoid the effect of concomitant diseases. Nonetheless, the ORs for symptoms, perceived poor health, cognitive function, and lower physical performance showed an increase in effect size with lower $FEV_1$. Importantly, while the ORs for mild to moderate $FEV_1$ categories were lower compared to severe $FEV_1$, the numbers of participants affected by these outcomes were considerably higher for these milder categories. This suggest that the burden associated with milder grades of low $FEV_1$ are high,

and their contribution to poor cognitive, physical, and functional outcomes may be underrecognized, since current practices would regard these $FEV_1$ levels as within the normal limits [22].

Last, we observed the strength of the association was particularly strong between $FEV_1$ with handgrip strength, standing balance, and moderate to severe respiratory symptoms. This is consistent with the growing body of literature, highlighting the association between reduced lung function with sarcopenia in the general populations [36,38,39]. Our findings add to this field by showing that reduced lung function is a part of the generalized manifestations of functional and cognitive decline and potentially frailty. Since lung function is an accessible and easily quantifiable measure, we propose that it may be an important indicator of general health and functional status irrespective of age, sex, smoking status, and underlying lung diseases. Its routine use in the community may lead to an increase in case finding and diagnosis of preclinical disability in the general population. Identifying these early and mild individuals will more likely offer greater opportunity for interventions and to modify their trajectory.

The strengths of this study include the large sample size and the representativeness of the general population. Data were collected using validated, standardized, and high-quality control methodology. The large number of covariates collected allowed for careful adjustments to reduce any confounding effects. The limitations include the cross-sectional analysis, which limits our ability to infer causality. The respiratory symptoms and perceived poor general health outcomes were self-reported and are subjected to recall bias. However, these questionnaires have been used in other epidemiological studies and have demonstrated robust associations with poor health outcomes and mortality [24,25,40]. The strict quality standards for spirometry measurements in CLSA may have selected mostly healthy individuals. However, the distribution in $FEV_1$ z-scores showed a slight skewness to the left with higher numbers of individuals with moderate to severe $FEV_1$ impairment (z-scores <-1 SD). Finally, these findings need to be examined in other populations from different ethnic and geographic backgrounds to confirm their generalizability.

In conclusion, we found a consistent and graded association between lower $FEV_1$ with higher odds of self-reported poor health, moderate to severe respiratory symptoms, and impaired cognitive performance in a large representative sample of the general population. Similar gradient of associations were observed for physical performance on validated tests, which have important prognostication for future functional impairment and poor health outcomes. Our findings suggest that low lung function may be an important and early finding of preclinical disability in the general population. There is also a high burden of moderate to severe respiratory symptoms and poor perceived health status even with very mild to moderate low $FEV_1$. Future studies are needed to examine the longitudinal associations between lower $FEV_1$ with future physical impairment, disability, and morbidity and whether strategies that promote lung health can improve the overall health trajectory with aging.

## Supporting information

**S1 Protocol. Protocol of planned analysis submitted to Hamilton Integrated Research Ethics Board.**
(DOCX)

**S1 STROBE Checklist. STROBE checklist.** STROBE, Strengthening the Reporting of Observational Studies in Epidemiology.
(DOCX)

**S1 Table. Baseline characteristics of the comprehensive and tracking cohorts.**
(DOCX)

**S2 Table. Contraindications to performing spirometry.**
(DOCX)

**S3 Table. Analyses on self-reported respiratory symptoms, self-perceived poor health status, and cognitive and physical performance for different grades of low FVC compared to reference (FVC > 0 SD) in the overall cohort and in participants without spirometry AO (shown here as $FEV_1/FVC >= LLN$).** AO, airflow obstruction; $FEV_1$, forced expiratory volume in 1 second; FVC, forced vital capacity; LLN, lower limit of normal.
(DOCX)

**S4 Table. Unadjusted stratified analysis by gender, smoking history, and baseline age for self-perceived poor health, respiratory symptoms, and low cognitive scores by grades of low $FEV_1$ relative to reference group ($FEV_1 > 0$ SD).** $FEV_1$, forced expiratory volume in 1 second; SD, standard deviation.
(DOCX)

**S5 Table. Adjusted stratified analyses by gender, smoking history, and baseline age for self-perceived poor health, respiratory symptoms, and low cognitive scores by grades of low $FEV_1$ relative to reference group ($FEV_1 > 0$ SD).** $FEV_1$, forced expiratory volume in 1 second; SD, standard deviation.
(DOCX)

**S6 Table. Unadjusted stratified analyses by gender, smoking history, and baseline age groups on physical performance according to grades of low $FEV_1$ relative to reference group ($FEV_1 > 0$ SD).** $FEV_1$, forced expiratory volume in 1 second; SD, standard deviation.
(DOCX)

**S7 Table. Adjusted stratified analyses by gender, smoking history, and baseline age groups on physical performance according to grades of lower $FEV_1$ z scores relative to reference group ($FEV_1 > 0$ SD).** $FEV_1$, forced expiratory volume in 1 second; SD, standard deviation.
(DOCX)

## Acknowledgments

This research has been conducted using the CLSA dataset Baseline Comprehensive version 3.0, under Application Number 161013.

## Author Contributions

**Conceptualization:** MyLinh Duong.

**Data curation:** MyLinh Duong, Ali Usman, Jinhui Ma, Michele Zaman, Alex Dragoman, Steven Jiatong Chen, Malik Farooqi.

**Formal analysis:** MyLinh Duong, Ali Usman, Jinhui Ma.

**Funding acquisition:** Parminder Raina.

**Methodology:** MyLinh Duong, Ali Usman, Jinhui Ma, Malik Farooqi.

**Project administration:** MyLinh Duong, Yangqing Xie, Julie Huang.

**Supervision:** MyLinh Duong, Julie Huang, Parminder Raina.

**Validation:** MyLinh Duong, Yangqing Xie, Julie Huang, Michele Zaman, Alex Dragoman, Steven Jiatong Chen, Malik Farooqi.

**Writing – original draft:** MyLinh Duong, Ali Usman.

**Writing – review & editing:** MyLinh Duong, Ali Usman, Jinhui Ma, Yangqing Xie, Julie Huang, Michele Zaman, Alex Dragoman, Steven Jiatong Chen, Malik Farooqi, Parminder Raina.

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
