## [Editor Report · Decision Letter 0]

4 Mar 2021

Dear Dr Duong, 

Thank you for submitting your manuscript entitled "Impact of FEV1  on Physical and Cognitive Health in the Canadian Longitudinal Study on Aging (CLSA)." for consideration by PLOS Medicine.

Your manuscript has now been evaluated by the PLOS Medicine editorial staff as well as by an academic editor with relevant expertise and I am writing to let you know that we would like to send your submission out for external peer review.

Kind regards,

Caitlin Moyer, Ph.D.

Associate Editor

PLOS Medicine

---

## [Editor Report · Decision Letter 1]

8 Mar 2021

Dear Dr Duong, 

Thank you for submitting your manuscript entitled "Impact of FEV1 on Physical and Cognitive Health in the Canadian Longitudinal Study on Aging (CLSA)." for consideration by PLOS Medicine.

Your manuscript has now been evaluated by the PLOS Medicine editorial staff as well as by an academic editor with relevant expertise and I am writing to let you know that we would like to send your submission out for external peer review.

Kind regards,

Caitlin Moyer, Ph.D.

Associate Editor

PLOS Medicine

---

## [Decision Letter · Decision Letter 2]

5 Sep 2021

Dear Dr. Duong,

Thank you very much for submitting your manuscript "Impact of FEV1 on Physical and Cognitive Health in the Canadian Longitudinal Study on Aging (CLSA)." (PMEDICINE-D-21-01015R2) for consideration at PLOS Medicine. 

Your paper was evaluated by a senior editor and discussed among all the editors here. It was also discussed with an academic editor with relevant expertise, and sent to four independent reviewers, including a statistical reviewer. The reviews are appended at the bottom of this email and any accompanying reviewer attachments can be seen via the link below:

[LINK]

In light of these reviews, I am afraid that we will not be able to accept the manuscript for publication in the journal in its current form, but we would like to consider a revised version that addresses the reviewers' and editors' comments. Obviously we cannot make any decision about publication until we have seen the revised manuscript and your response, and we plan to seek re-review by one or more of the reviewers. 

We expect to receive your revised manuscript by Sep 24 2021 11:59PM. Please email us (plosmedicine@plos.org) if you have any questions or concerns.

We look forward to receiving your revised manuscript. 

Sincerely,

Caitlin Moyer, Ph.D.

Associate Editor 

PLOS Medicine

plosmedicine.org

From the Academic Editor:

1. I agree with one of the reviewers that this is an association study. The use of 'effect' suggests causality. This should be attended to.

2. The focus is on FEV1. it would be nice to get some more information on the standardisation procedures for its measurement. This is a notorious measure in the field, especially if it is being done by several different research assistants (which is likely to be the case). There are some pre-requirements - not having smoked for an hour, not after a large meal, avoid alcohol, etc. It is best done in the office, but field measurement may have been the only option. When compared to normative data, they had 25% <1 SD and 6% <2 SD, which suggests a sample skewed to the left (although this is a large population-based study). This does not negate the results, but is worthy of discussion.

3. There are several measures of frailty that are being linked to cognition and to each other. A multivariate regression model may be instructive to see if FEV1 has any predictive value over and above the other measures? Is it a unique association, or is it that since frailty is known to be linked to cognition, and the measures of frailty are correlated, and we are seeing this in a univariate analysis.

4. Do they have any other measure of general health? Many studies do use self-report on one question, but it would be nice to have something more comprehensive and/or objective.

Additional editorial requests:

5. Please revise your title according to PLOS Medicine's style. Your title must be nondeclarative and not a question. It should begin with main concept if possible. "Effect of" should be used only if causality can be inferred, i.e., for an RCT. Please place the study design ("A randomized controlled trial," "A retrospective study," "A modelling study," etc.) in the subtitle (ie, after a colon).

6. Financial disclosure: Please describe the sources of funding for the study, in addition to noting that The funders had no role in study design, data collection and analysis, decision to

publish, or preparation of the manuscript.

7. Data availability statement: The Data Availability Statement (DAS) requires revision. PLOS Medicine requires that the de-identified data underlying the specific results in a published article be made available, without restrictions on access, in a public repository or as Supporting Information at the time of article publication, provided it is legal and ethical to do so. Please see the policy at

http://journals.plos.org/plosmedicine/s/data-availability

and FAQs at

http://journals.plos.org/plosmedicine/s/data-availability#loc-faqs-for-data-policy

For each data source used in your study:

8. Throughout: Please include line numbers with the revised version.

9. Abstract: Please structure your abstract using the PLOS Medicine headings (Background, Methods and Findings, Conclusions). Please combine the Methods and Findings sections into one section, “Methods and findings”.

10. Abstract: Background: Abstract Background: The final sentence should clearly state the study question.

11. Abstract: Methods and Findings: * Please include the study design, population and setting, number of participants, years during which the study took place, length of follow up, and main outcome measures. * Please quantify the main results (with 95% CIs and p values). * In the last sentence of the Abstract Methods and Findings section, please describe the main limitation(s) of the study's methodology.

12. Abstract: Conclusions: * Please address the study implications without overreaching what can be concluded from the data; the phrase "In this study, we observed ..." may be useful.

13. Author summary: At this stage, we ask that you include a short, non-technical Author Summary of your research to make findings accessible to a wide audience that includes both scientists and non-scientists. The Author Summary should immediately follow the Abstract in your revised manuscript. This text is subject to editorial change and should be distinct from the scientific abstract. Please see our author guidelines for more information: https://journals.plos.org/plosmedicine/s/revising-your-manuscript#loc-author-summary

14. Throughout: For in-text reference citations, please place reference numbers in square brackets before the sentence punctuation, like this [1]. For multiple references, please do not include spaces within brackets.

15. Methods: Please provide the name(s) of the institutional review board(s) that provided ethical approval for the study, and please specify whether informed consent of participants was written or oral.

16. Methods: Please provide more clarification on the selection of/ any differences between those selected for physical/cognitive/clinical assessments vs the tracking cohort. “Selected participants

were interviewed and had standardized physical, cognitive and clinical assessments

(comprehensive cohort) to provide data on demographics, lifestyle, health, and clinical

information. In the remaining participants (tracking cohort, n=21,241), similar data was collected

by a telephone interview”

17. Methods: Please clarify: “Participants screened positive for major contra-indications were

Excluded.”

18. Methods: Please remove the trademark symbol from (Tracker Freedom®

Wireless)

19. Methods: Did your study have a prospective protocol or analysis plan? Please state this (either way) early in the Methods section.

20. Methods: Please ensure that the study is reported according to the STROBE guideline, and include the completed STROBE checklist as Supporting Information. Please add the following statement, or similar, to the Methods: "This study is reported as per the Strengthening the Reporting of Observational Studies in Epidemiology (STROBE) guideline (S1 Checklist)."

21. Results: Throughout, please present the main results in the text, with 95% CIs and p values. For the assessment of respiratory, health, cognitive status with FEV1, please report both the adjusted and unadjusted analyses (with 95% CIs and p values).

22. Results: Please provide quantitative results for these findings (reported with 95% CIs and p values): “The proportion of obstructive impairment (whether defined by FEV1/FVC <0.70 or <LLN), increased with lower FEV1; reaching approximately 50% in the severely low FEV1 group. The distribution of age, sex, and height across the FEV1 categories were similar. However, lower FEV1 was associated with lower physical activity, lower education level (primary school or lower); and higher rates of current smoking, obesity (BMI >30 kg/m2), asthma, COPD, CVD and multiple co-morbidities.”

23. Discussion: Please present and organize the Discussion as follows: a short, clear summary of the article's findings; what the study adds to existing research and where and why the results may differ from previous research; strengths and limitations of the study; implications and next steps for research, clinical practice, and/or public policy; one-paragraph conclusion.

24. Discussion: We suggest tempering this statement, as a formal dose-response analysis did not seem to be reported: “The novel finding here, is the highly consistent dose-response relationship between lower FEV1 with higher prevalence and ORs of self-reported respiratory symptoms…”

25. Table 2 and Table 3: Please provide the exact p values rather than p<0.05, etc. Please report p<0.001 where applicable. Please also provide the results from unadjusted analyses.

26. Figure 1-2: Please provide exact p values, or indicate in the legend that these are reported elsewhere. Please also include the results from unadjusted analyses.

27. Appendix 1 and Appendix 2: Please report exact p values, except for p<0.001, where applicable. Please also provide results from unadjusted analyses.

Comments from the reviewers:

Reviewer #1: This study examines cross-sectional baseline data for associations between FEV1 with self-reported respiratory symptoms, perceived poor health, cognitive and physical performance. 

Comments:

Can the authors please assess whether the comprehensive cohort with complete baseline data (22,822 participants) is representative of the wider nationally representative CLSA (51,338 participants)? 

"Self-reported data from questionnaires included: age (45-54, 55-64, 65-74 & 75+ years); sex; smoking status (never [lifetime <100 cigarettes], former [last cigarette smoked >12 months], and current); education (below secondary, secondary and >secondary); known cardiovascular disease (CVD) (angina, congestive heart failure, myocardial infarction), chronic obstructive pulmonary disease (COPD), asthma and major chronic diseases (incorporated into the co-morbidity index 0, 1-2, >3). Height and weight were measured with standardized methods and equipment. Body mass index (BMI) was calculated as weight divided by height-squared and categorized into <25, 25-30, >30 kg/m2 . Self-reported physical activity was assessed by the Physical Activity Scale for the Elderly (PASE) questionnaire with higher weighted scores indicating higher activity levels in the previous 7 days."

Did the authors consider treating age and BMI as continuous variables (if available in this format from the questionnaires)?

Can the authors please acknowledge the potential bias of self-reporting, for both covariates and outcomes (e.g. perceived general health) in the study limitations? 

"The semantic fluency test assessed cognitive performance by asking participants to name as many animals within 60 seconds. Test scores were standardized for age, sex and education; and scores <45 considered impaired".

Did the authors consider treating cognitive assessment (i.e. semantic fluency test results) as a continuous variable? 

Did the authors complete any sensitivity analyses on the cut-off of <45?

"Means and frequencies (%) were used to describe the sample. Hierarchical logistic regression was used to estimate the effects of low FEV1 (relative to FEV1>0sd as reference), adjusted for age, sex,

BMI, education, smoking status, self-reported asthma, COPD, CVD, co-morbidity index (excluding asthma, COPD and CVD). Centers were treated as random effect. The goodness-of-fit tests (likelihood ratio test, deviance, AIC, BIC criteria), multi-collinearity (tolerance and variance inflation factor) and visual inspection of residuals were performed to assess model stability and robustness. Trimmed inflation and analytical (rescaled) weights were applied; to reduce the effect of selection bias and maintain the national representativeness and generalizability of the data. Similar analyses after removing participants with spirometric obstruction (FEV1/FVC<LLN) were performed to ensure our findings were not confounded by diagnosed and undiagnosed COPD." 

Did the authors examine the distribution of data, in order to confirm if mean or median may be more appropriate?

The authors should state in the methods section that they computed SD for continuous variables (as shown in Table 1).

Furthermore, the authors have completed suitable multilevel linear regression models to estimate the differences between each level of low FEV1 z-score relative to reference (FEV1%>0sd), but have not mentioned this in the methods section.

"The prevalence and adjusted odds ratios (ORs) for these three outcomes showed a graded increase with lower FEV1 (Figure 1). The relationship was particularly strong (steep) between worsening FEV1 and increasing odds of moderate-severe respiratory symptoms, after accounting for differences in demographic and clinical factors. While the ORs were highest for the severely low FEV1 category; the absolute numbers of participants affected by these outcomes were far higher for the mild and moderately low FEV1 categories than for the severely low FEV1 group. "

Can the authors please quote accompanying relevant statistics in the main text, alongside each stated study finding?

Table 1: Can the authors please complete statistical tests comparing baseline differences between groups, and present the results within this table?

Overall, the authors have appropriately applied multilevel modelling, suitably assessing model stability and exploring multi-collinearity. Selection bias has been somewhat accounted for, and a thorough sensitivity analysis has been undertaken. The authors rightly acknowledge that the cross-sectional analysis limits the ability to infer causality, and that generalisability is restricted requiring findings to be examined in other populations.

Reviewer #2: Many epidemiologic studies have demonstrated that poor lung function, as indicated by a low forced expiratory volume in 1s (FEV1) and forced vital capacity (FVC), is associated with an increased risk of adverse outcomes, cardiovascular or pulmonary. The FEV1 indicator is included in frailty models. The reviewed article adds new information about its association with self reported symptoms, cognitive and physical performance in the general population. The researchers found a consistent and graded association between lower FEV1 with higher odds of self-reported poor health, moderate-severe respiratory symptoms and impaired cognitive performance, in a large representative sample of the general population. This results add that reduced lung function is a part of the generalized manifestations of functional and cognitive decline and potentially frailty. The results are important to the researches in geriatric, lung functions, cognitive disorders. There are the implications for patient care and public policy. 

In general the research is of very high methodological quality. The sample size is sufficient to prove the hypothesis. The data and analyses fully support the claims. The strength of the study is that large representative sample of the Canadian population aged 45 to 85 years living independently in the community was investigated. The large number of covariates collected allowed for careful adjustments to reduce any confounding effects. The figures and tables are correct and of high technical quality. The manuscript is well organized and written clearly enough to be accessible to non-specialists.

I do not see any needs of major or minor improvement. 

Reviewer #3: This is a cohort of a population-based Canadian about lung function and the associated with higher mortality and adverse cardiopulmonary outcomes.The author analyse 22,822 adults (52% females, mean age 58.8 [sd 9.6]); the association between forced expiratory volume in 1 second (FEV1) with self-reported moderate to severe respiratory symptoms, perceived poor health, cognitive and physical performance were examined; using multilevel regression adjusted for potential confounders. The study is weel desging and the methdods and statitics sounds correct. The authors concluded that there is a consistent and graded association between lower FEV1 with higher odds of self-reported poor health, moderate-severe respiratory symptoms and impaired cognitive performance, in a large representative sample of the general population. Similar gradient of associations were observed for physical performance on validated tests, which have important prognostication for future functional impairment and poor health outcomes.

I have seen on the system three version of the article, I have read the R2 and I have any important consideration. There is no ethical and methodological issues. The introduction sounds good as well discussion and conclusioin that is machet with the objetctive of the article.

Reviewer #4: This study addresses an interesting association between low FEV1 and self reported "present health". The findings are of interest but should be qualified. Firstly the authors report an association and all implications of causality should be deleted. Secondly the participants were asked to "rate their present health" this is not the same as health status as measured by specific instruments. It is not clear why one of the validated measures of health status was not used and tis should be discussed. It would also be interesting to know if the findings also held true for the FVC measurements, as this would support the concept that the finding either reflects poor lung development or poor muscle strength rather than being related to airflow obstruction - the authors should present these data as well and discuss the results.

[LINK]

---

## [Decision Letter · Decision Letter 3]

23 Dec 2021

Dear Dr. Duong,

Thank you very much for re-submitting your manuscript "Associations between lung function with physical and cognitive health in the Canadian Longitudinal Study on Aging (CLSA), a multi-center population-based observational cohort study." (PMEDICINE-D-21-01015R3) for review by PLOS Medicine.

I have discussed the paper with my colleagues and the academic editor and it was also seen again by two reviewers. I am pleased to say that provided the remaining editorial and production issues are dealt with we are planning to accept the paper for publication in the journal.

[LINK]

We expect to receive your revised manuscript within 2 weeks. Please email us (plosmedicine@plos.org) if you have any questions or concerns.

We look forward to receiving the revised manuscript by Jan 06 2022 11:59PM.   

Sincerely,

Caitlin Moyer, Ph.D.

Associate Editor 

PLOS Medicine

plosmedicine.org

Requests from Editors:

1. Financial disclosure: Please clarify the funding statement associated with your manuscript. Youn note both that “There was no funding support for the current analysis and work in this report.” but also list a number of funding sources as well as that “The funders had no role in study design, data collection and analysis, decision to publish, or preparation of this manuscript.”

Please address this discrepancy. This information should describe sources of funding that have supported the work included in this submission.The statement should include:Specific grant numbers, Initials of authors who received each award, and URLs to sponsors’ websites. Also state whether any sponsors or funders (other than the named authors) played any role in: Study design, Data collection and analysis, Decision to publish, Preparation of the manuscript. If they had no role in the research, include this sentence: “The funders had no role in study design, data collection and analysis, decision to publish, or preparation of the manuscript.” If the study was unfunded, include this sentence as the Financial Disclosure statement: “The author(s) received no specific funding for this work."

2. Data availability statement: Please update this section of the manuscript submission system with the details provided in the Data Access section of the main text (and please remove this section from the main text).

3. Throughout: Please include line numbers running continuously throughout the text of the manuscript.

4. Abstract: Background We suggest revising to “...forced expiratory volume in 1 second (FEV1), an indicator of lung function, with broad markers of general health…”

5. Abstract: Methods and Findings: Please clarify what is meant by “Compared to the Global Lung Function Initiative predictive values” (could this be written as: “Based on Global Lung Function Initiative reference values”?).

6. Abstract: Methods and Findings: For the OR provided for perceived poor health and impaired cognitive performance, please indicate which of the three levels is being described for each (mild, moderate, severely low). Please also note that these are the adjusted OR, and please mention the factors adjusted for.

7. Abstract: Methods and Findings, final sentence: Please revise to make the study limitations more clear: “A limitation is that the study is observational and causality cannot be inferred.” or similar.

8. Abstract: Conclusions: First sentence: Please remove the word “robust” from the sentence.

Author summary: Why was this study done? We suggest combining the first two bullet points.

9. Funding: Please remove the “Funding” section from the main text of the manuscript. Please ensure that all information is accurately entered into the manuscript submission system “Financial disclosure” section.

10. Data Access: As mentioned above, please remove the “Data Access” section from the main text of the manuscript. Please ensure that all information is completely and accurately entered in the manuscript submission system in the “Data availability” section.

11. Results: Please clarify this sentence “Compared to the overall cohort, the proportion of individuals with airflow obstruction whether defined by the GLI FEV1/FVC <LLN criterion (11% in overall vs. 58%), or FEV1/FVC<0.70 (5.4% vs. 45%) were higher in the severe FEV1 category.” We suggest noting as “11% in overall vs 58% in the severe FEV1 category” for example, if this is accurate.

12. Results: Sensitivity analyses: Please note the location where the data are shown for the analyses where participants with airflow obstruction were removed (e.g. Table 2 and Table 3).

13. Discussion: First paragraph: There is a typo in the sentence: “Furthermore, they were These associations were highly consistent across different age groups…”

14. Table 1: It would be helpful to provide a label at the top to indicate that the column headers indicate z score categories for GLI-FEV1 measures (this applies to all tables presented with z score column headers). Please define the abbreviation “FEV”

15. Table 2 and Table 3: Please define the abbreviations for FFV, LLN, FVC in the legend.

16. Figure 1: Please note in the legend the p values are reported in Table 2. Please provide a visual key within the figure indicating the definition of the open/closed circles and squares.

17. Figure 2: Please note in the legend the p values are reported in Table 3.Please provide a visual key within the figure indicating the definition of the open/closed circles and squares.

18. References: Please remove the italicized fonts from the reference list. Please check the formatting of each reference. Please note that there should be punctuation following the journal title. Please use the "Vancouver" style for reference formatting, and see our website for other reference guidelines https://journals.plos.org/plosmedicine/s/submission-guidelines#loc-references

Please use the same guidelines for formatting the references listed in the appendix.

19. STROBE Checklist: Thank you for including the STROBE Checklist. Please revise to indicate the locations within the text using Section and Paragraph Numbers (e.g. Methods, paragraph 1). Please do not report page numbers or line numbers.

20. Supporting Information files: List supporting information captions at the end of the manuscript file. Do not submit captions in a separate file. The file number and name are required in a caption, and we highly recommend including a one-line title as well. You may also include a legend in your caption, but it is not required. Authors may use almost any description as the item name for a supporting information file as long as it contains an “S” and number. For example, “S1 Appendix” and “S2 Appendix,” “S1 Table” and “S2 Table,” and so forth.

21. Appendix 1: We suggest including the protocol as a separate file, and we suggest renaming to “S1_Protocol” or similar.

22. Appendix 2: In the legend, it would be helpful to briefly define what is meant by “comprehensive” and “tracking” cohort.

23. Appendix 4: Please note that under “Cognitive Impairment” the categories for Smokers vs Non-smokers are repeated, and it seems the male/female stratification is missing.

Comments from Reviewers:

Reviewer #1: The authors have satisfactorily considered and responded to each comment in turn, and have amended the manuscript accordingly.

Reviewer #4: The authors have addressed my concerns

[LINK]

---

## [Editor Report · Decision Letter 4]

10 Jan 2022

Dear Dr Duong, 

On behalf of my colleagues and the Academic Editor, Perminder Singh Sachdev, I am pleased to inform you that we have agreed to publish your manuscript "Associations between lung function with physical and cognitive health in the Canadian Longitudinal Study on Aging (CLSA), a multi-center population-based observational cohort study." (PMEDICINE-D-21-01015R4) in PLOS Medicine.

Please also address the following points:

-Title: Please revise the title (and please update this in the submission system as well as on the manuscript) to: "Associations between lung function with physical and cognitive health in the Canadian Longitudinal Study on Aging (CLSA): A cross-sectional study from a multi-center national cohort"

-Acknowledgement: Please move the “Acknowledgements” section to follow the Discussion section.

-Abstract: Methods and Findings: Please revise to “58% females, mean age 58.8 years [standard deviation (sd) 9.6]”. 

-Abstract: Methods and Findings: Please revise the final sentence of this section to: “A limitation of the current study is the observational nature of these findings and that causality cannot be inferred.”

-Abstract: Conclusions: Please revise to: “These findings support the broader implications…”

-Author summary: “What do these findings mean” Please revise to” …and cognitive and physical outcomes in the general population.”

-Methods: Line 61: Please verify if “ndd” should be “NDD”.

-STROBE Checklist: Please do not refer to page numbers in the checklist. Please revise the checklist, using only section and paragraph numbers to refer to locations in the text (e.g. Methods, paragraph 1).

PRESS

Sincerely, 

Caitlin Moyer, Ph.D. 

Associate Editor 

PLOS Medicine